# Prognostic Value of the Neutrophil-to-Lymphocyte Ratio before and after Radiotherapy for Anaplastic Thyroid Carcinoma

**DOI:** 10.3390/cancers13081913

**Published:** 2021-04-15

**Authors:** Jiyun Park, Jun Park, Jung Hee Shin, Young-Lyun Oh, Hyun-Ae Jung, Man-Ki Chung, Jun-Ho Choe, Yong-Chan Ahn, Sun-Wook Kim, Jae-Hoon Chung, Tae-Hyuk Kim, Jae-Myoung Noh

**Affiliations:** 1Division of Endocrinology and Metabolism, Department of Medicine, Samsung Medical Center, Sungkyunkwan University School of Medicine, Seoul 06351, Korea; nove1123.park@samsung.com (J.P.); pjun113@gmail.com (J.P.); swkimmd@skku.edu (S.-W.K.); thyroid@skku.edu (J.-H.C.); 2Department of Radiology and Center for Imaging Science, Thyroid Center, Samsung Medical Center, Sungkyunkwan University School of Medicine, Seoul 06351, Korea; helena35.shin@samsung.com; 3Department of Pathology, Thyroid Center, Samsung Medical Center, Sungkyunkwan University School of Medicine, Seoul 06351, Korea; yl.oh@samsung.com; 4Division of Hematology-Oncology, Department of Medicine, Samsung Medical Center, Sungkyunkwan University School of Medicine, Seoul 06351, Korea; hyunae.jung@samsung.com; 5Department of Otorhinolaryngology-Head and Neck Surgery, Samsung Medical Center, Sungkyunkwan University School of Medicine, Seoul 06351, Korea; manki.chung@samsung.com; 6Division of Breast and Endocrine Surgery, Department of Surgery, Samsung Medical Center, Sungkyunkwan University School of Medicine, Seoul 06351, Korea; junho.choe@samsung.com; 7Department of Radiation Oncology, Samsung Medical Center, Sungkyunkwan University School of Medicine, Seoul 06351, Korea; ycahn.ahn@samsung.com

**Keywords:** anaplastic thyroid carcinoma, neutrophil-to-lymphocyte ratio, neutrophil, lymphocyte, radiotherapy

## Abstract

**Simple Summary:**

Systemic hematologic markers such as the neutrophil–lymphocyte ratio (NLR) are attracting attention in a simple blood test predicting survival and treatment outcomes in various solid cancers. In our study including 40 patients with anaplastic thyroid cancer (ATC) who underwent radiotherapy, we found that high NLR before and after radiotherapy was associated with poor survival. This study result means that host immunity has an important role in patients with ATC, which can be important information for designing future investigations on the impact of immune system modulating therapy in ATC.

**Abstract:**

The neutrophil–lymphocyte ratio (NLR) is a marker of systemic inflammation, and its elevation has recently been associated with poor survival in many solid cancers. Leukocyte elevation and lymphocyte reduction are associated with a poor response to radiotherapy (RT). This study aimed to assess the prognostic value of NLR before and after RT for anaplastic thyroid carcinoma (ATC). This retrospective study analyzed 40 patients with ATC who received RT with available complete blood cell count data from November 1995 through May 2020 at Samsung Medical Center (Seoul, Korea). Patients were classified into two groups according to the NLR before and after RT. The median overall survival (OS) was 8.9 months (range, 3.5–18.2) in the low NLR group (<3.47) and 5.2 months (range, 2.7–7.5) months in the high NLR group (≥3.47). The association between NLR and OS was also observed in multivariable Cox regression analysis (hazard ratio, 3.18; 95% confidence interval, 1.15–8.85; *p* = 0.026). The OS curves differed significantly according to post-RT NLR (*p* = 0.036). A high NLR before and after RT may be significantly associated with poor OS in patients with ATC who receive RT.

## 1. Introduction

Anaplastic thyroid carcinoma (ATC) is an aggressive form of thyroid cancer that is associated with a poor prognosis. The management of patients with ATC is very difficult, and there is no effective therapy; however, multimodal therapy (MMT) is commonly recommended [1,2,3].

Several studies have reported that systemic hematological markers can predict poor outcomes in some malignancies. In particular, the baseline neutrophil-to-lymphocyte ratio (NLR), which represents the ratio of peripheral circulating neutrophil and lymphocyte counts, is a marker of inflammation, and increased NLR has been shown to be an indicator of a poor prognosis in many cancers, including that of the breast, colon, and head and neck [4,5,6]. With regard to thyroid cancers, it is known that a high baseline NLR is associated with increased tumor size, metastasis, and disease-free survival in patients with differentiated thyroid cancer [7,8,9]. Moreover, NLR can be used to discriminate aggressive forms of thyroid cancer from well differentiated cancer [10]. 

Recently, not only the baseline NLR but also NLR before and after treatments such as chemoradiotherapy or immunotherapy was reported to be associated with survival in several solid tumors [4,11,12,13,14]. Leukocytosis is associated with resistance to radiotherapy (RT). On the other hand, lymphocytes are sensitive to RT and exhibit antitumor effects [4,15]. Furthermore, RT can lead to lymphopenia and reduced lymphocyte function [14]. Considering the interaction between changes in hematological values and RT, it can be hypothesized that NLR before and after RT can play a prognostic role in the evaluation of treatment outcomes in several cancers. However, to our knowledge, no previous studies have investigated the prognostic value of NLR before and after RT for ATC. 

The primary aim of this study was to assess the prognostic impact of NLR before and after RT in patients with ATC.

## 2. Materials and Methods

### 2.1. Study Population

From November 1995 through May 2020, 120 patients diagnosed with ATC were treated at Samsung Medical Center, Seoul, Republic of Korea. Among these patients, 64 received RT. The exclusion criteria were as follows: previously differentiated carcinoma (*n* = 5), anaplastic change <5% (*n* = 1), unavailable baseline complete blood cell count (CBC) and differential count data (*n* = 17), and palliative surgery after definitive concurrent chemoradiotherapy (*n* = 1). A total of 40 patients were included, 19 of whom underwent surgery and RT with or without systemic treatment such as tyrosine kinase inhibitor (TKI) therapy or cytotoxic chemotherapy. The other 21 patients received only RT with or without systemic treatment. Figure 1 shows the study flowchart. The study was approved by the Institutional Review Board of Samsung Medical Center (SMC-IRB 2020-10-033).

### 2.2. Laboratory and Pathological Data

Patient and tumor characteristics and CBC data were extracted from the electronic medical records. Only CBC data available for counting absolute lymphocytes and absolute neutrophils were used. NLR was calculated as the absolute neutrophil count (ANC) divided by the absolute lymphocyte count (ALC). In the group who underwent surgery, baseline CBC data were obtained on a date nearest to surgery date when the interval between surgery and RT was within 100 days. However, for one patient with an interval longer than 100 days, data were obtained on a date closest to the RT start date. In the group that did not undergo surgery, baseline CBC data were obtained on a date nearest to the RT start date. Post-RT CBC data were obtained within 4–6 weeks after the last day of RT. The American Joint Committee on Cancer (AJCC) staging system (8th edition) was used for tumor staging [16].

### 2.3. Treatment and Primary Outcome

Surgical interventions included total thyroidectomy with or without bilateral neck dissection as per the surgeon’s decision. For cytotoxic chemotherapy, a regimen including doxorubicin, cisplatin, and paclitaxel was administered before surgery or as a form of concurrent chemoradiotherapy. For patients treated with TKIs, multikinase inhibitors such as lenvatinib and selective *BRAF* inhibitors such as dabrafenib plus trametinib or vemurafenib were administered when the patients had *BRAF600E* mutation. 

RT was delivered with 4-, 6-, or 10-MV photons generated from a linear accelerator. Thirteen (32.5%) and 20 (50.0%) patients received three-dimensional conformal RT (3D-CRT) and intensity-modulated RT (IMRT), respectively. The remaining seven (17.5) patients received two-dimensional RT, and most of them were treated before 2002. For 3D-CRT and IMRT, the gross tumor volume (GTV) was defined as the volume of the primary tumor and involved lymph node(s) on the basis of all available clinical information. The clinical target volume (CTV) was delineated by adding 3–5 mm margins in all directions from the GTV, and the margins were optionally modified in accordance with the anatomical boundaries. In the case of postoperative RT, the CTV included the primary tumor bed and pathologically involved regional lymphatics with adequate margins. The radiation dose fractionation was based on the aim of RT and the patient’s performance status. For example, 30 Gy in 10 fractions was prescribed for eight patients requiring palliative RT, while 59.4 Gy in 27 fractions was prescribed for eight patients requiring postoperative RT. 

MMT was defined as a combination of two or more of the following treatments: surgery, cytotoxic chemotherapy, TKI therapy, and RT. The primary outcome was overall survival (OS), which was calculated as the time from the date of diagnosis to the occurrence of death from any cause.

### 2.4. Statistical Analysis

All data are presented as medians with interquartile ranges (IQRs) for continuous variables and numbers with percentages for categorical variables. The t-test and Wilcoxon rank-sum test were used to compare continuous variables depending on whether a normal distribution was satisfied. The chi-square test was used for categorical variables. Area under the curve (AUC) value using receiver operating characteristic (ROC) analysis was performed to evaluate the optimal cutoff value for NLR. Kaplan–Meier analysis and the log-rank test were used to compare survival between the two groups. Hazard ratios (HRs) and 95% confidence intervals (CIs) for OS were calculated by univariate and multivariate analyses using the multivariable Cox proportional hazard regression method. Variables with a *p*-value ≤0.2 in univariate analysis were included in multivariate analysis. All *p*-values were two-tailed, and *p*-values of <0.05 were considered statistically significant. All statistical analyses were performed using STATA version 16.0 (StataCorp, College Station, TX, USA).

## 3. Results

### 3.1. Patient Characteristics

Table 1 shows the baseline characteristics according to the use of surgical resection. The median age at diagnosis was 65.1 years (IQR 57.8–72.2) in the surgery group and 70.8 years (IQR 58.1–78.6) in the non-surgery group. The median tumor size was 4.8 cm (IQR 3.1–5.4) in the surgery group and 5.3 cm (IQR 4.2–6.5) cm in the non-surgery group. The stage IVC disease was higher in the group without surgery than in the group with surgery (71.4% vs. 31.6%, *p* = 0.012), while the patients who received a total RT dose of EQD2_10_ (equivalent dose in 2-Gy fractions with an α/β ratio of 10) of 60 Gy or more was significantly higher in the group with surgery (*n* = 12 (63.1%) with surgery vs. *n* = 2 (9.5%) without surgery, *p* = 0.012). The frequency of systemic treatment, including cytotoxic chemotherapy and TKI therapy, was not different between the two groups (*p* = 0.273 and *p* = 0.201, respectively). There were significantly more patients with an NLR of ≥3.47 in the group without surgery than in the group with surgery (*n* = 12 (57.1%) without surgery vs. *n* = 3 (15.8%) with surgery, *p* = 0.007). 

### 3.2. Optimal Cutoff Value for the NLR and Cinicopathological Characteristics According to NLR

The median and IQR of NLR in the total population were 2.87 (1.50–5.28). To evaluate the optimal cutoff value for NLR, the ROC curve for OS was used. The optimal cutoff value for the baseline NLR was set to 3.47, with an area under the curve (AUC) of 0.64 (sensitivity 48.4, specificity 88.9, Appendix A). The optimal cutoff value for NLR after RT was 3.44, with an AUC of 0.65 (sensitivity 76.5, specificity 62.5, Appendix A). According to these results, the patients were classified into two groups (high NLR group, NLR ≥3.47, *n* = 15 (37.5%); low NLR group, NLR <3.47, *n* = 25 (62.5%)).

The clinicopathological characteristics of patients according to NLR are shown in Table 2. The NLR was 1.99 (1.28–2.67) in the low NLR group and 6.33 (4.59–11.11) in the high LNR group (*p* < 0.001). The tumor size and stage IVC disease were not different between the two groups (4.5 cm (IQR 3.3–5.5) in the low NLR group vs. 5.4 cm (IQR 4.4–6.5) in the high NLR group, *p* = 0.145; *n* = 11 (44%) in the low NLR group vs. *n* = 10 (66.7%) in the high NLR group, *p* = 0.165). Surgical resection was higher in the low NLR group (64.0% vs. 20.0%, *p* = 0.007). 

### 3.3. Overall Survival According to the Baseline NLR

Figure 2 shows the Kaplan–Meier curves for OS. The high NLR was associated with poorer survival (*p* = 0.002). The median OS was 8.9 months (IQR 3.5–18.2) in the low NLR group and 5.2 months (IQR 2.7–7.5) in the high NLR group. Stage IVC disease, surgery, MMT, a high total radiation dose, and a high NLR were associated with OS in univariate Cox regression analysis (Table 3). However, only high NLR was associated with OS in multivariate Cox regression analysis for the total population (HR: 3.18, 95% CI: 1.15–8.85, *p* = 0.026; Table 3).

### 3.4. Subgroup Analysis in the Group with and without Surgery

Figure 3 shows the Kaplan–Meier curves for OS in the surgery and non-surgery groups (*p* = 0.017 vs. *p* = 0.376, respectively). A high NLR was associated with poor survival in the group with surgery, but not in the group without surgery. In univariate Cox regression analysis for the surgery group, a high total radiation dose and high NLR were associated with OS. However, in multivariate analysis, only a high total radiation dose remained a significant factor (high total radiation dose, HR: 0.05, 95% CI: 0.005–0.43, *p* = 0.007; NLR, HR: 2.07, 95% CI: 0.28–15.39, *p* = 0.478; Table 4). In the group without surgery, stage IVC and TKI therapy were associated with OS in univariate Cox analysis, and both of them remained as significant factors in multivariate Cox analysis (Stage IVC, HR: 9.37, 95% CI: 2.22–39.46, *p* = 0.002; TKI, HR: 0.14, 95% CI: 0.04–0.51, *p* = 0.003; Table 4).

### 3.5. Association between NLR after RT and OS

In total, 25 patients with available CBC data after RT were included in this analysis. The median and IQR of post-RT-NLR were 6.60 (2.75–9.81). Ten patients had low post-RT NLR, and 15 patients had high post-RT NLR. Figure 4 shows the Kaplan–Meier curves for OS according to the post-RT-NLR. High post-RT-NLR was associated with poor OS (*p* = 0.036). The median OS was 13.9 (IQR, 3.5–34.3) and 4.3 (IQR, 2.5–7.5) months in the low and high post-RT-NLR groups, respectively. In univariate Cox analysis, the HR (95% CI) for high post-RT-NLR was 2.98 (1.03–8.68) (*p* = 0.045).

When 25 patients were divided into four groups according to the change in NLR after RT (low–low, low–high, high–low, and high–high), there were no statistically significant differences in OS among the four groups (*p* = 0.051; Figure 5). However, the HR (95% CI) for the high–high NLR group versus the low–low NLR group was 6.40 (1.55–26.40) in univariate Cox analysis (*p* = 0.01).

## 4. Discussion

In this study, we evaluated whether NLR before and after RT for ATC was associated with OS. The high baseline NLR was significantly associated with poorer OS in both univariate and multivariate analyses for the total population. The subgroup analysis showed that the high NLR group had poor OS in the patients who underwent surgery. In this subgroup with surgery, high NLR lost its significant association with OS in multivariate Cox analysis; however, the high total radiation dose remained significant. Because only three patients had high NLR, we could not conduct further stratified analysis for evaluating the interaction between NLR and the radiation dose. The high post-RT NLR was also associated with poor OS. Only 25 out of 40 patients had available CBC data after RT, and 14 patients of them underwent surgery while 11 did not undergo surgery. A univariate Cox analysis was attempted to confirm the association between RT dose and OS in patients with post-RT-NLR value, but this was not possible because the number of surviving patients in the group with a total radiation dose of <60 Gy was quite small (Appendix A). 

Although high NLR has been reported to be associated with poor survival in many solid cancers, the association between ATC and NLR has not been reported. The role of the inflammatory cells in the development and progression of thyroid cancer has been discussed in previous studies [17,18,19]. To date, it is known that chronic lymphocytic thyroiditis in PTC induced improved prognosis, with fewer lymphocytes in patients with poorly differentiated thyroid cancer (PDTC) and ATC than in PTC patients, suggesting lymphocytes have a protective role in thyroid cancer [18]. Monocytes accelerate tumor progression and suppress normal immune responses, and a low lymphocyte-to-monocyte ratio is associated with poor survival in ATC [15,17]. Neutrophils secrete highly reactive metabolites and eliminate pathogens. However, these metabolites can induce mutation or proliferation of the cancer cells [18]. Our study showed that high NLR was associated with poor survival before and after RT in ATC, which is consistent with results in previous studies considering the role of each inflammatory cell. 

As we mentioned in the introduction, it was reported that high NLR was associated with increased tumor size, metastasis, and disease-free survival in patients with differentiated thyroid cancer [7,8]. As the tumor size and staging increase, higher amounts of tumor-derived or tumor-induced factors are likely to be released into the periphery of cancer cells, thus increasing the amount of circulating neutrophils [20]. In our study, tumor size was not associated with NLR. PTC is an indolent form of cancer that grows slowly, and the NLR value is also lower compared to ATC [10]. This means inflammation may be less important in the initiation of carcinogenesis in differentiated cancer, with a lower effect on systemic inflammation [8]. The NLR value may increase as the size increases, gradually provoking systemic inflammation in PTC. On the other hand, ATC is aggressive and grows fast, inducing inflammatory processes in the early phase of carcinogenesis. Therefore, NLR was thought not to be associated with tumor size. Furthermore, one study reported that NLR can discriminate aggressive forms of thyroid cancer from well differentiated cancer [10]. In this previous study, the NLR of PTC, with a median value of 1.57 (range, 0.28–16.29), could discriminate PDTC and ATC from PTC, and the cut-off value of LNR for discriminating PDTC from ATC was 3.8. The NLR of our study population was 2.87 (range, 0.94–22.51), a higher value compared to PTC patients in a previous study. However, only 14 out of 40 in our study population had NLR values equal to or higher than 3.8. When the cut off value was set to 3.8, the sensitivity was 45.16% and specificity was 89% for death in our study population. This means that even in ATC patients, NLR greater than 3.8 represents high mortality. Considering PDTC is as aggressive as ATC, LNR greater than 3.8 can discriminate aggressive types of thyroid cancer and poor survival. Studies on larger patient cohorts are warranted to define a uniform NLR cut-off value for thyroid cancer. 

High NLR is also associated with poorer outcomes in patients with immune checkpoint inhibitors in other solid cancers [21]. Circulating neutrophil counts are associated with tumor infiltrating neutrophils [22]. Therefore, decreased circulating lymphocytes are associated with reduced lymphocyte in tumor tissue, resulting in less chance to interact with immune check-point inhibitors. Further intervention studies are needed to confirm that NLR can predict the response of immunotherapy in thyroid cancer.

Although our study population included patients who received RT, surgery, MMT, and a high total radiation dose were associated with better survival, as reported in previous studies [3,23]. In particular, our subgroup analysis showed that the total radiation dose was a good prognostic factor for OS in patients who received surgery, while TKI treatment was a good prognostic factor for patients who received RT without surgery. This means that a high-dose adjuvant RT for patients receiving surgery and adjuvant TKI therapy for patients receiving RT without surgery could improve OS in patients with ATC. A previous study reported that a high radiation dose resulted in better survival, even in patients with unresected ATC [24]. In our study population, more than 90% of the patients without surgery received low-dose RT. Therefore, there was a limitation in evaluating the association between the radiation dose and OS in the group without surgery.

Several studies have reported that selective BRAF inhibitors or multikinase inhibitors are effective in patients with ATC who have previously undergone surgery or RT [25,26]. In our study, only three patients in the surgery group received TKI treatment, and TKI treatment was not associated with OS in this group. However, in the group without surgery, TKI treatment was performed in more than thirty percent of the patients and was also associated with better OS, as shown in previous studies [24,25]. Further prospective studies are warranted to validate this association. 

This study has some limitations. First, it was a small, single-center, retrospective study; therefore, there are some inherent limitations such as selection bias and validation. Second, CBC data were not available for all patients, and patients’ medical condition or situation at the time of CBC measurement was not known. NLR is a readily available and affordable biomarker; however, it can be affected by unmeasurable medical conditions; therefore, NLR values have to be interpreted cautiously in a clinical context. Third, we could not perform additional immunohistochemical staining to find tumor-associated neutrophils or flow cytometry to evaluate the lymphocyte subtype due to a lack of available tissue specimens. Therefore, the difference in neutrophil and lymphocyte infiltration in ATC between low NLR and high NLR groups was not evaluated. However, we focused on the association between blood NLR and OS of the patients with ATC based on the assumption that blood NLR could reflect tumor tissue NLR. Further studies are needed to confirm the consistency of NLR in blood and tissue. 

## 5. Conclusions

High NLR was associated with poor survival in patients with ATC who received RT. Furthermore, high post-RT NLR was also associated with poor survival. This study evaluated the role of NLR as a readily available and affordable blood marker for predicting survival in ATC patients before and after RT. Prospective studies are needed to verify the association between NLR and OS before and after RT in patients with ATC.

## Figures and Tables

**Figure 1 cancers-13-01913-f001:**
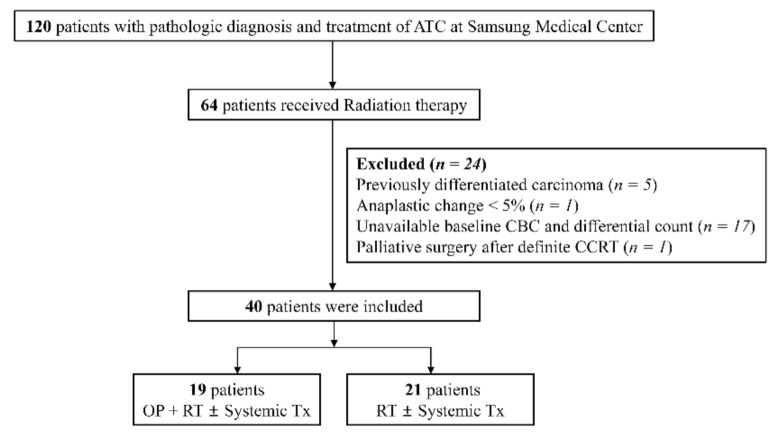
Study flowchart. ATC, anaplastic thyroid carcinoma; OP, operation; CBC, complete blood count; CCRT, concur rent chemoradiotherapy; Tx, treatment.

**Figure 2 cancers-13-01913-f002:**
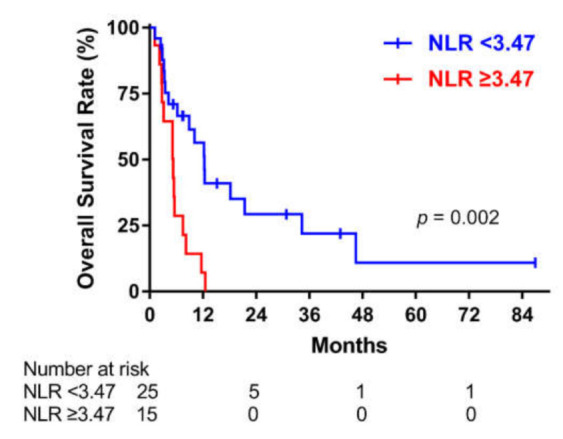
Overall survival according to the baseline neutrophil-to-lymphocyte ratio (NLR) in the overall sample of patients who underwent radiotherapy for anaplastic thyroid carcinoma.

**Figure 3 cancers-13-01913-f003:**
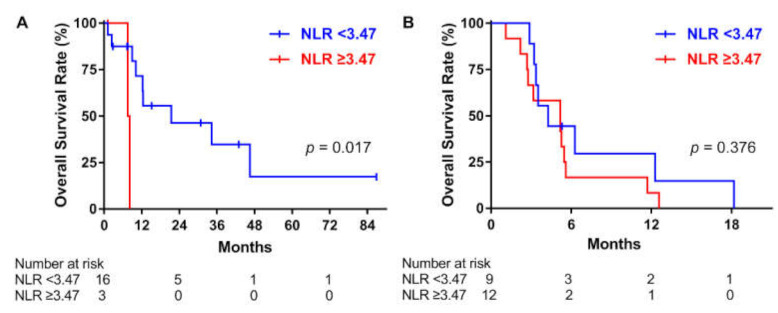
Overall survival according to the neutrophil-to-lymphocyte ratio (NLR) in patients who underwent radiotherapy (RT) with or without surgery for anaplastic thyroid carcinoma. (**A**) Operation + RT ± systemic treatment. (**B**) RT ± systemic treatment.

**Figure 4 cancers-13-01913-f004:**
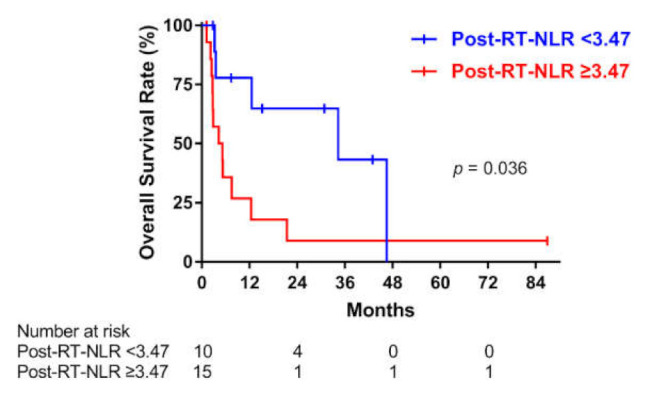
Overall survival according to the post-radiation therapy (post-RT) neutrophil-to-lymphocyte ratio (NLR) in patients with anaplastic thyroid carcinoma.

**Figure 5 cancers-13-01913-f005:**
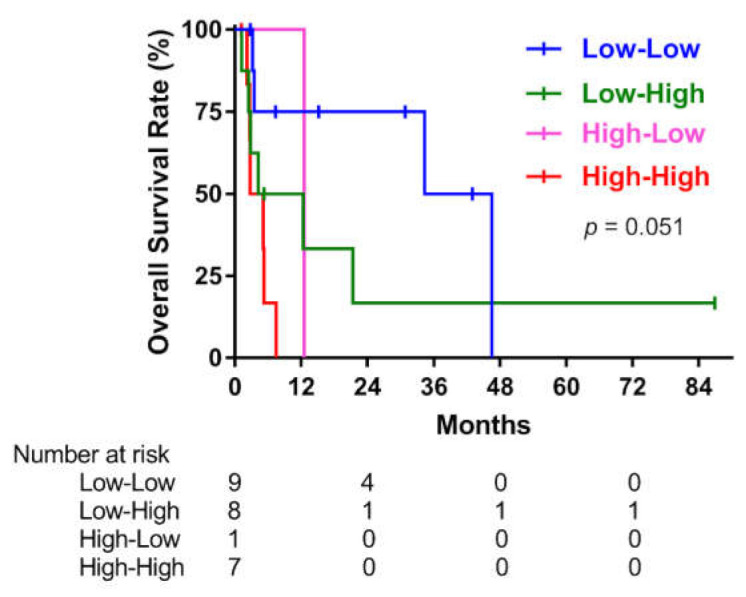
Overall survival according to the change in the neutrophil-to-lymphocyte ratio (NLR) after radiotherapy for anaplastic thyroid carcinoma.

**Table 1 cancers-13-01913-t001:** Baseline characteristics of patients who received RT with or without surgery for anaplastic thyroid carcinoma.

Variables	OP + RT ± Systemic Tx (*n* = 19)	RT ± Systemic Tx (*n* = 21)	*p*-Value
Age (years)	65.1 (57.8–72.2)	70.8 (58.1–78.6)	0.383
Sex			0.220
Male	9 (47.4)	6 (28.6)	-
Female	10 (52.6)	15 (71.4)	-
Tumor size (cm)	4.8 (3.1–5.4)	5.3 (4.2–6.5)	0.245
Stage			0.012
IVA & IVB	13 (68.4)	6 (28.6)	-
IVC	6 (31.6)	15 (71.4)	
Total dose (EQD2_10,_ Gy)			<0.001
<60.0	7 (36.9)	19 (90.5)	-
≥60.0	12 (63.1)	2 (9.5)	-
Systemic treatment			-
Cytotoxic chemotherapy	5 (26.32)	9 (42.86)	0.273
TKI	3 (15.8)	7 (33.3)	0.201
Multimodal therapy	19 (100)	13 (61.9)	0.003
Baseline ALC (µL)	2120 (1740–2363)	1810 (1310–2380)	0.173
Baseline ANC (µL)	4220 (2970–5803)	6598 (5814–11,729)	0.017
Baseline NLR	2.24 (1.4–3.1)	4.95 (2.54–6.60)	0.013
Baseline NLR (≥3.47)	3 (15.8)	12 (57.1)	0.007

Continuous variables are presented as medians (interquartile ranges). Categorical variables are presented as numbers (percentages). ALC, absolute lymphocyte count; ANC, absolute leukocyte count; EQD2_10_, equivalent dose in 2-Gy fractions with an α/β ratio of 10; NLR, neutrophil-to-lymphocyte ratio; RT, radiotherapy; OP, operation; TKI, tyrosine kinase inhibitor; Tx, treatment.

**Table 2 cancers-13-01913-t002:** Clinicopathological characteristics according to the baseline NLR in patients with anaplastic thyroid carcinoma treated by radiotherapy.

Variables	Low NLR (*n* = 25)	High NLR (*n* = 15)	*p*-Value
Age (years)	67.4 (63.1–72.2)	67.3 (57.9–73.0)	0.882
Sex			0.354
Male	8 (32.0)	7 (53.3)	-
Female	17 (68.0)	8 (46.7)	-
Tumor size (cm)	4.5 (3.3–5.5)	5.4 (4.4–6.5)	0.145
Stage			0.165
IVA & IVB	14 (56.0)	5 (33.3)	-
IVC	11 (44.0)	10 (66.7)	-
Total dose (EQD2_10,_ Gy)			0.123
<60.0	14 (56.0)	12 (80.0)	-
≥60.0	11 (44.0)	3 (20.0)	-
Surgery	16 (64.0)	3 (20.0)	0.007
Systemic treatment			
Cytotoxic chemotherapy	7 (28.0)	7 (46.7)	0.231
TKI	6 (24.0)	4 (26.7)	0.850
Multimodal therapy	20 (80.0)	12 (80.0)	1.000
Baseline NLR	1.99 (1.28–2.67)	6.33 (4.95–11.11)	<0.001
Baseline ALC (µL)	2218 (1890–2460)	1480 (1222–1853)	<0.001
Baseline ANC (µL)	4188 (2680–5802)	10,494 (7677–14,060)	<0.001

Continuous variables are presented as medians (interquartile ranges). Categorical variables are presented as numbers (percentages). ALC, absolute lymphocyte count; ANC, absolute leukocyte count; EQD2_10_, equivalent dose in 2-Gy fractions with an α/β ratio of 10; NLR, neutrophil-to-lymphocyte ratio; TKI, tyrosine kinase inhibitor.

**Table 3 cancers-13-01913-t003:** Cox proportional hazard model for all-cause mortality in the overall sample of patients with anaplastic thyroid carcinoma treated by radiotherapy.

Variables	Univariate Analysis		Multivariate Analysis	
	HR (95% CI)	*p*-Value	HR (95% CI)	*p*-Value
Age (years)	1.00 (0.97–1.04)	0.819	-	-
Sex (male)	1.18 (0.56–2.48)	0.661	-	-
Size (cm)	1.23 (0.99–1.52)	0.059	0.86 (0.64–1.16)	0.328
Stage (IVC)	3.59 (1.59–8.07)	0.002	2.97 (0.81–10.95)	0.102
Surgery	0.21 (0.09–0.50)	<0.001	0.70 (0.24–2.03)	0.507
Systemic treatment				
Cytotoxic CTx	1.57 (0.74–3.33)	0.238	-	-
TKI	0.76 (0.33–1.72)	0.506	-	-
Multimodal therapy	0.36 (0.16–0.84)	0.018	0.42 (0.10–1.77)	0.236
EQD2_10_(≥60 Gy)	0.10 (0.03–0.31)	<0.001	0.27 (0.06–1.22)	0.089
NLR (≥3.47)	3.32 (1.51–7.29)	0.003	3.18 (1.15–8.85)	0.026

CI, confidence interval; CTx, chemotherapy; EQD2_10_, equivalent dose in 2-Gy fractions with an α/β ratio of 10; HR, hazard ratio; NLR, neutrophil-to-lymphocyte ratio; TKI, tyrosine kinase inhibitor.

**Table 4 cancers-13-01913-t004:** Cox proportional hazard model for all-cause mortality in patients who received RT with or without surgery for anaplastic thyroid carcinoma.

Treatment	OP + RT ± Systemic Tx (*n* = 19)				RT ± Systemic Tx (*n*= 21)			
Variables	Univariate Analysis		Multivariate Analysis		Univariate Analysis		Multivariate Analysis	
	HR (95% CI)	*p*-Value	HR (95% CI)	*p*-Value	HR (95% CI)	*p*-Value	HR (95% CI)	*p*-Value
Age (years)	1.00 (0.93–1.09)	0.831	-	-	0.98 (0.94–1.02)	0.375	-	-
Sex (male)	1.74 (0.52–5.77)	0.369	-	-	1.79 (0.60–5.39)	0.298	-	-
Size (cm)	1.03 (0.67–1.58)	0.893	-	-	1.14 (0.88–1.46)	0.322	-	-
Stage (IVC)	2.21 (0.52–9.37)	0.281	-	-	2.96 (0.95–9.22)	0.061	9.37 (2.22–39.46)	0.002
Systemic Tx	-	-	-	-	0.88 (0.34–2.25)	0.787	-	-
Cytotoxic CTx	1.11 (0.29–4.30)	0.883	-	-	1.86 (0.66–5.19)	0.238	-	-
TKI	0.74 (0.15–3.52)	0.702	-	-	0.44 (0.16–1.17)	0.099	0.14 (0.04–0.51)	0.003
EQD2_10_ (≥60 Gy)	0.04 (0.005–0.36)	0.004	0.05 (0.005–0.43)	0.007	0.46 (0.10–2.04)	0.304	-	-
NLR (≥3.47)	7.76 (1.08–55.92)	0.042	2.07 (0.28–15.39)	0.478	1.52 (0.59–3.90)	0.382	-	-

CI, confidence interval; CTx, chemotherapy; EQD2_10_, equivalent dose in 2-Gy fractions with an α/β ratio of 10; HR, hazard ratio; NLR, neutrophil-to-lymphocyte ratio; OP, operation; RT, radiotherapy; TKI, tyrosine kinase inhibitor; Tx, treatment.

## Data Availability

The data presented in this study are available within the article and Appendix A.

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
