# Peer review of "Prognostic Value of the Neutrophil-to-Lymphocyte Ratio before and after Radiotherapy for Anaplastic Thyroid Carcinoma"

_cancers, 2021, doi:10.3390/cancers13081913_

Round 1

Reviewer 1 Report

In literature is reported that increased levels of Neutrophil-to-lymphocyte ratio (NLR) are correlated with poor prognosis in many types of human cancers. Moreover, it is already reported by other group that high circulating level of NLR correlates with tumor size in patients affected by differentiated thyroid cancer. However, the potential prognostic role of NLT in anaplastic thyroid cancer (ATC) remains unknown.

The manuscript by Park et al., examines the expression of NLR in ATC patients before and after radiotherapy. The authors provide data that high expression of NLR was associated with poor survival in 40 ATC patients.

The study was well designed and performed; the data appear to be very encouraging. Although the results are interesting, there are some issues that should be addressed.

Major concerns are:

1) The major weakness is: the few tumor tissues used for the study of the expression of NLR. It is true that ATC is, fortunately, very rare form of human cancer; but on the based on these considerations, I recommend to interrogate publicly available thyroid cancer datasets. These online datasets can be used to analyze the NLR expression in a higher group of ATC patients.

2) It would be interesting to analyze NLR expression in The Cancer Genome Atlas (TCGA) datasets using tools like cbioportal, firebrowse. With this regard, the author could search the expression of NLR also in PTC series, considering prognosis-associated factors (for instance classic PTCs versus tall-cell variant PTCs). Finally, the authors could also look for the correlation between NLR and overall survival in these datasets.

3) What is the expression level of NLR protein in ATC tissues? It is possible obtain this information by performing immunohistochemical analyzes in your data set? Literature could help you about these points.

4) Furthermore, what is the expression of NLR in the plasma of healthy subjects? This point could improve the study supporting the data presented in the manuscript.

Regarding the data shown in the manuscript I have some minor comments.     Minor concerns are:

1) In “Laboratory and pathological data” paragraph (line 85) the authors should be more meticulous regarding the material and methods. In particular more detail should be provided about how NLP levels were calculated. What about absolute neutrophil count (ANC) and the absolute lymphocyte count (ALC)?

2) Concerning statical analysis, it might be interesting to show the image it concerns Area under the curve (AUC) value, used to evaluate the optimal cutoff value for NLR.

3) Considering the literature, the authors should consider their findings in the context of a previous reports demonstrating that NLR correlates with tumor size also in patients with differentiated thyroid cancer

4) How do you explain the data reported by Cho and colleagues (Reference number 9) concerning that hypothesis that NLR could discriminate ATC versus poorly or well differentiated cancer.

Author Response

We appreciated the excellent and valuable comments.

We did our best to make a satisfactory revision to your requests. 

We used "track changes" in Microsoft Word and the page in response letter is version using the track change function. 

We attach the response letter and hope that our revised manuscript is satisfactory for you. 

Thank you. 

Reviewer: 1

Comments to the Author

In literature is reported that increased levels of Neutrophil-to-lymphocyte ratio (NLR) are correlated with poor prognosis in many types of human cancers. Moreover, it is already reported by other group that high circulating level of NLR correlates with tumor size in patients affected by differentiated thyroid cancer. However, the potential prognostic role of NLT in anaplastic thyroid cancer (ATC) remains unknown.

The manuscript by Park et al., examines the expression of NLR in ATC patients before and after radiotherapy. The authors provide data that high expression of NLR was associated with poor survival in 40 ATC patients.

The study was well designed and performed; the data appear to be very encouraging. Although the results are interesting, there are some issues that should be addressed.

Major concerns are:

1) The major weakness is: the few tumor tissues used for the study of the expression of NLR. It is true that ATC is, fortunately, very rare form of human cancer; but on the based on these considerations, I recommend to interrogate publicly available thyroid cancer datasets. These online datasets can be used to analyze the NLR expression in a higher group of ATC patients.

Answer] Thank you very much for the thoughtful comment. NLR is the ratio of blood circulating neutrophil and lymphocyte ratio. Circulating neutrophil and lymphocytes can migrate into tumor tissue and infiltration of specific leukocyte subsets in tumor is associated with prognosis. A high density of tumor associated neutrophil (TAN) is associated with poor prognosis in several solid cancer (Mukaida al.). It is known that lymphocytes and dendritic cells are markedly reduced in undifferentiated or poorly differentiated thyroid cancer (Ugolini et al.). NLR is not the concept of expression of specific “NLR gene” and this is not a protein. Our study did not evaluate the expression of NLR of tumor tissue, instead, evaluated blood circulating LNR based on the assumption that blood NLR could reflect tumor tissue NLR. Therefore, it is not possible to find out NLR expression in tumor tissue. However, it might be possible to find TAN in tumor tissue using immunohistochemical methods or flow cytometry to evaluate lymphocyte subtype. However, this is retrospective study and there was not available tissue specimen to perform additional immunohistochemical stating or flow cytometry. We added this point to limitation section.   

[Discussion, 7th paragraph, page 11]

Third, we could not perform additional immunohistochemical staining to find tumor associated neutrophil or flow cytometry to evaluate lymphocyte subtype due to lack of available tissue specimen. Therefore, the difference of neutrophil and lymphocyte infiltration in ATC between low NLR and high NLR groups were not evaluated. However, we focused the association between blood NLR and OS of the patients with ATC based on the assumption that blood NLR could reflect tumor tissue NLR. Further studies are needed to confirm the consistency of NLR in blood and tissue.

* Mukaida et al.

Mukaida, N.; Sasaki, S.I.; Baba, T. Two-Faced Roles of Tumor-Associated Neutrophils in Cancer Development and Progression. International journal of molecular sciences 2020, 21.

* Ugolini et al.

Ugolini, C.; Basolo, F.; Proietti, A.; Vitti, P.; Elisei, R.; Miccoli, P.; Toniolo, A. Lymphocyte and immature dendritic cell infiltrates in differentiated, poorly differentiated, and undifferentiated thyroid carcinoma. Thyroid : official journal of the American Thyroid Association 2007, 17, 389-393.

2) It would be interesting to analyze NLR expression in The Cancer Genome Atlas (TCGA) datasets using tools like cbioportal, firebrowse. With this regard, the author could search the expression of NLR also in PTC series, considering prognosis-associated factors (for instance classic PTCs versus tall-cell variant PTCs). Finally, the authors could also look for the correlation between NLR and overall survival in these datasets.

Answer] The authors do appreciate the reviewer’s valuable comments. As we mentioned above, NLR is not a single gene and therefore we could not perform expression analysis using public datasets. We search “NLR gene” in The Cancer Genome Atlas (TCGA) using cBioPortal and Firebrowse Broad GDAC. However, we could not find it. As you expected, tall-cell variant PTC had poorer prognosis compared to classic PTC, it would be interesting whether there is NLR difference according to pathologic category within PTC in future study.

3) What is the expression level of NLR protein in ATC tissues? It is possible obtain this information by performing immunohistochemical analyzes in your data set? Literature could help you about these points.

 Answer] The authors do appreciate the reviewer’s comments. We could not evaluate the LNR expression level because it is not a single protein but a ratio of blood circulating leukocyte. As the blood circulating neutrophil and lymphocyte increases, the leukocytes of tissue also increase, therefore, it might be possible to find neutrophil and lymphocyte in tumor tissue using immunohistochemical analyzes. However, this study focused on the blood marker instead of tissue neutrophil. We have reflected the reviewer’s important point that blood leukocyte counts are associate d with tissue leukocyte count in the discussion section.

[Discussion, 4th paragraph, page 11]

High NLR is also associated with poorer outcome in the patient with immune checkpoint inhibitor in other solid cancer [21]. Circulating neutrophil counts are associated with tumor infiltrating neutrophil [22]. Therefore, decreased lymphocyte are associated with reduced lymphocyte in tumor tissue less chance to interact with immune check point inhibitor. Further intervention studies are needed to confirm that NLR can predict the response of immunotherapy in thyroid cancer.

[Reference section]

  1. Moses, K.; Brandau, S. Human neutrophils: Their role in cancer and relation to myeloid-derived suppressor cells. Seminars in immunology 2016, 28, 187-196.
  2. Sacdalan, D.B.; Lucero, J.A.; Sacdalan, D.L. Prognostic utility of baseline neutrophil-to-lymphocyte ratio in patients receiving immune checkpoint inhibitors: a review and meta-analysis. OncoTargets and therapy 2018, 11, 955-965.

4)Furthermore, what is the expression of NLR in the plasma of healthy subjects? This point could improve the study supporting the data presented in the manuscript.

Answer] The authors do appreciate the reviewer’s valuable comments. We did not find NLR expression in healthy patients as the same reason described above. Instead, we searched studies that evaluated the blood NLR in healthy subjects. A cross sectional study to evaluated NLR of 2212 healthy population in Iran reported the mean and standard deviation of NLR was 1.70 ± 0.70 and there was no significant difference between sex, age group (Moosazadeh et al.). And in Belgium, 413 healthy patients showed mean NLR 1.65 ± 1.96 (Forget et al.). Most of the studies evaluating NLR in cancer used their cut-off data using AUC of ROC curve and did not use the cut-off value of normal control. There is no absolute cut off value to distinguish between healthy patients and cancer patients. NLR is readily available and affordable biomarker, however, can be affected by unmeasurable medical condition such as nutrition and inflammation. Therefore, NLR have to be interpreted cautiously in a clinical context.

[Discussion, 7th paragraph, page 11]

This study has some limitations. First, it was a small, single-center, retrospective study; therefore, there are some inherent limitations such as selection bias and validation. Second, CBC data were not available for all patients and patient’s medical condition or situation at the time of CBC measurement cannot be known. NLR is readily available and affordable biomarker, however, can be affected by unmeasurable medical condition, therefore, NLR have to be interpreted cautiously in a clinical context.

* Moosazadeh et al.

Moosazadeh, M.; Maleki, I.; Alizadeh-Navaei, R.; Kheradmand, M.; Hedayatizadeh-Omran, A.; Shamshirian, A.; Barzegar, A. Normal values of neutrophil-to-lymphocyte ratio, lymphocyte-to-monocyte ratio and platelet-to-lymphocyte ratio among Iranian population: Results of Tabari cohort. Caspian journal of internal medicine 2019, 10, 320-325.

*Forget et al.

Forget, P.; Khalifa, C.; Defour, J.P.; Latinne, D.; Van Pel, M.C.; De Kock, M. What is the normal value of the neutrophil-to-lymphocyte ratio? BMC research notes 2017, 10, 12.

Regarding the data shown in the manuscript I have some minor comments.   

Minor concerns are:

1) In “Laboratory and pathological data” paragraph (line 85) the authors should be more meticulous regarding the material and methods. In particular more detail should be provided about how NLP levels were calculated. What about absolute neutrophil count (ANC) and the absolute lymphocyte count (ALC)?

Answer] Answer] Thank you very much for the thoughtful comment. CBC data shows the differential cell count, which can determine the percentage of leukocytes including lymphocytes, monocytes, and neutrophils. This is called the absolute lymphocyte count and the absolute neutrophil count. NLR is calculated from the CBC as the absolute lymphocyte count divided by the absolute monocyte count. I added further descriptions to the “Laboratory and pathological data” section in response to the reviewer’s comment.

[Method, 2.2. Laboratory and pathological data section, page 2]:

Patient and tumor characteristics and CBC data were extracted from the electronic medical records. Only CBC data available for counting absolute lymphocytes and absolute neutrophils were used. NLR was calculated as the absolute neutrophil count (ANC) divided by the absolute lymphocyte count (ALC).

2) Concerning statical analysis, it might be interesting to show the image it concerns Area under the curve (AUC) value, used to evaluate the optimal cutoff value for NLR.

Answer] Thank you for the thoughtful comment. We agree with the reviewer’s opinion. We added ROC curve to supplemental material due to lack of main manuscript space.

[Result, 3.2. Optimal cutoff value for the NLR and clinicopathological characteristics according to NLR, page 5]:

The optimal cutoff value for the baseline NLR was set to 3.47, with an area under the curve (AUC) of 0.64 (sensitivity 48.4, specificity 88.9, Figure S1). The optimal cutoff value for NLR after RT was 3.44, with an AUC of 0.65 (sensitivity 76.5, specificity 62.5, Figure S2).

Figure S1. ROC curve assessing the cutoff points of baseline NLR.

ROC, receiver operating characteristics; NLR, neutrophil-to-lymphocyte ratio

Figure S2. ROC curve assessing the cutoff points of post-RT-NLR.

ROC, receiver operating characteristics; RT, radiation therapy; NLR, neutrophil-to-lymphocyte ratio

3) Considering the literature, the authors should consider their findings in the context of a previous reports demonstrating that NLR correlates with tumor size also in patients with differentiated thyroid cancer

Answer] We do appreciate the thoughtful comments. We examined two articles about association between NLR and clinicopathologic characteristics of well differentiated thyroid cancer (DTC). Liu’s paper (Reference number 11) divided NLR values of 159 patients into tertile. As the tertile increased, tumor size increased. Lymph node invasion or staging was not different according to tertile of NLR. Manatakis’s paper (Reference number 8) evaluated the difference of NLR according to clinicopathologic characteristic. Bilateral, multifocal, and lymph node positive PTC had higher NLR. However, NLR was not different according to tumor size. Studies examining NLR in well-differentiated thyroid cancer were heterogeneous in statistical methodology and varied largely in their sample sizes, therefore, produced inconsistent results. Manatakis’s paper also reported, in general, higher NLR was associated larger tumor size, multifocality, lymph node metastasis and staging although their data could not show consistent results with previous studies. Summarizing these results, one meta-analysis reported that tumor size and metastasis were associate with NLR (Feng’s paper, Reference number 9). In our study, tumor size was not associated with NLR. PTC is indolent form of cancer, grows slowly and NLR value is also lower compared to ATC (Cho’s paper, Reference 10). This means inflammation may be less important in the initiation of carcinogenesis in differentiated cancer, less affecting systemic inflammation (Manatakis’s paper, Reference number 8). The NLR may increase as the size increases provoking systemic inflammation gradually in PTC. On the other hand, ATC is aggressive and grows fast inducing inflammatory process in early phase of carcinogenesis. Therefore, NLR is thought to be not associated with tumor size.

[Introduction, 2nd paragraph, page 2]

With regard to thyroid cancers, it is known that a high baseline NLR is associated with increased tumor size, metastasis and disease free survival and a poor prognosis in patients with differentiated thyroid cancer [7-9].

[Discussion, 2nd and 3rd paragraph, page 10]

Although high NLR have been reported to be associated with poor survival in many solid cancer, the association between ATC and NLR has not been reported yet. The role of the inflammatory cells in the development and progression of thyroid cancer have been discussed in previous studies [17-19]. To date, it is known that chronic lymphocytic thyroiditis in PTC induce improved prognosis and lymphocytes are less in the patients with poorly differentiated thyroid cancer (PDTC) and ATC than in PTC patients suggesting lymphocyte have protective role in thyroid cancer [18]. The monocytes accelerate tumor progression and suppress normal immune system and low lymphocyte-to-monocyte ratio are associated with poor survival in ATC [15,17]. The neutrophil secrets highly reactive metabolites and eliminate pathogen. However, these metabolites can induce mutation or proliferation of the cancer cell [18]. Our study showed high NLR was associated with poor survival before and after RT in ATC, which is consistent results in the previous studies considering the role of each inflammatory cells.

As we mentioned in introduction, it was reported that high NLR was associated with increased tumor size, metastasis and disease free survival in patients with differentiated thyroid cancer [7,8]. As the tumor size and staging increases, higher amounts of tumor derived or tumor induced factors are likely to be released into the periphery of cancer thus increasing the circulating neutrophil amount [20]. In our study, tumor size was not associated with NLR. PTC is indolent form of cancer, grows slowly and NLR value is also lower compared to ATC [10]. This means inflammation may be less important in the initiation of carcinogenesis in differentiated cancer, less affecting systemic inflammation [8]. The NLR may increase as the size increases provoking systemic inflammation gradually in PTC. On the other hand, ATC is aggressive and grows fast inducing inflammatory process in early phase of carcinogenesis. Therefore, NLR was thought not to be associated with tumor size.

[Reference]

  1. Feng, J.; Wang, Y.; Shan, G.; Gao, L. Clinical and prognostic value of neutrophil-lymphocyte ratio for patients with thyroid cancer: A meta-analysis. Medicine 2020, 99, e19686.

4) How do you explain the data reported by Cho and colleagues (Reference number 9) concerning that hypothesis that NLR could discriminate ATC versus poorly or well differentiated cancer.

Answer] The authors do appreciate the reviewer’s valuable comments. Cho’s paper evaluated the capability of the neutrophil to lymphocyte ratio (NLR) as a diagnostic tool to discriminate between poorly differentiated thyroid cancer (PDTC) and anaplastic thyroid cancer (ATC) from well differentiated thyroid cancer (WDTC) such as papillary thyroid cancer (PTC). The median NLR of 3870 patients with thyroid nodule was 1.57 (range, 0.28-16.29) and they were divided into 1935 patients on each groups. NLR was 2.13 (range, 1.57-16.29) in high NLR group and 1.21 (range, 0.28-1.57) in low NLR group. The LNR of the patients with 14 PDTC and 7 ATC was1.87 (range, 1.17-3.71) and 3.81 (range, 1.19-14.07) respectively. 18/21 of those patients had high NLR. The NLR of PTC, 1.57 (range, 0.28-16.29) could discriminate PDTC and ATC from PTC and the cut-off value for discriminating PDTC from ATC was 3.8. In our study, the NLR in total population was 2.87 (0.94-22.51). Since our study includes more than 5 times more ATC patients compared to Cho’s study, there may be some differences, the NLR value of our study population can be considered to have high NLR value enough to distinguish PTC. 14/40 patients had NLR over 3.8 in our population. When the cut off value was set to 3.8, the sensitivity was 45.16% and specificity was 89% for death in our total study population. This means even in ATC patients, NLR over 3.8 represents high mortality. And considering PDTC is as aggressive as ATC, LNR over 3.8 can be considered as aggressive type of thyroid cancer. Studies in larger patient cohorts are needed to define uniform cut-off value within thyroid cancer.

[Introduction, page 2]

Moreover, NLR can be used to discriminate aggressive forms of thyroid cancer from well differentiated cancer [10].

[Discussion, 3rd paragraph, page 10]

Furthermore, one study reported that NLR can discriminate aggressive forms of thyroid cancer from well differentiated cancer [10]. In this previous study, NLR of PTC, median 1.57 (range, 0.28-16.29) could discriminate PDTC and ATC from PTC and the cut-off value of LNR for discriminating PDTC from ATC was 3.8. Considering the NLR of our study population was 2.87 (range, 0.94-22.51), it is higher value compared to PTC patients in previous study. However, only 14 out of 40 our study population had NLR value same or over than 3.8. When the cut off value was set to 3.8, the sensitivity was 45.16% and specificity was 89% for death in our study population. This means even in ATC patients, NLR over 3.8 represents high mortality. Considering PDTC is as aggressive as ATC, LNR over 3.8 can discriminate aggressive type of thyroid cancer and poor survival. Studies in larger patient cohorts are warranted to define uniform NLR cut-off value within thyroid cancer.

Thank you very much for the helpful comments and suggestions. We did our best to address the reviewer’s requests and hope that it would be the satisfied version for the publication.

Reviewer 2 Report

Dear Authors,
“ Prognostic Value of the Neutrophil-to-Lymphocyte Ratio Before and After Radiotherapy for Anaplastic Thyroid Carcinoma” .
Below you can find my comments.
  1. The aim of the article is clear – it is yet another step on the way to widen our knowledge about risk factors in the prognosis of anaplastic thyroid carcinoma in an adult population.
  2. The introduction: The present state of knowledge is sufficiently presented, though in a very simple way.
  3. The methods are simple but conducted and presented in the right way.
  4. The summary is in line with previously presented data and their accuracy and interpretation. The most relevant reports are cited.

Final Comments:
The manuscript presents an interesting issue concerning the prognostic value of anaplastic thyroid carcinoma based on a simple hematological parameters.

Author Response

Thank you very much for finding out the strengths of our research and giving us a valuable comments. 

Reviewer: 2

Dear Authors,

“ Prognostic Value of the Neutrophil-to-Lymphocyte Ratio Before and After Radiotherapy for Anaplastic Thyroid Carcinoma”

Below you can find my comments.

The aim of the article is clear – it is yet another step on the way to widen our knowledge about risk factors in the prognosis of anaplastic thyroid carcinoma in an adult population.

The introduction: The present state of knowledge is sufficiently presented, though in a very simple way.

The methods are simple but conducted and presented in the right way.

The summary is in line with previously presented data and their accuracy and interpretation. The most relevant reports are cited.

Final Comments:

The manuscript presents an interesting issue concerning the prognostic value of anaplastic thyroid carcinoma based on a simple hematological parameters.

Answer] Thank you very much for finding out the strengths of our research and giving us a valuable comment.

Reviewer 3 Report

This article deals with an interesting topic evaluating the prognostic value of the neutrophil-to-lymphocyte ratio(NLR) before and after radiotherapy for anaplastic thyroid cancer. Because NLR has been reported as an important systemic hematologic marker that could be used in the prediction of the outcome of various cancers, and it is not applied to very aggressive and fatal anaplastic thyroid cancer. While there have been reported that higher NLR is associated with increased tumor size and aggressiveness in differentiated thyroid cancer.

  1. Overall, the description of results and discussion is inconsistent and very confusing. Authors have tried to demonstrate that higher NRL group has poorer outcome in general regardless of applying different therapeutic options. They showed that significantly improved overall survival in those with lower NLR (<3.47) than higher NLR group for both pre- and post-therapy data in a total group. (pre-therapy in Figure 2 and post-therapy, in Figure 4). In subgroup analysis, this was only confirmed in the group those underwent radiotherapy after surgery only. They showed that changes of NLR after RT did not have a statistically significant impact on the prediction of overall survival.
  2. Most importantly, the authors stated that higher baseline NLR was associated with improved survival (page 5 line 181-182) and even in the discussion section (page 9. Line 243-245). What happened? Higher NLR Authors described completely oppositely from those data and repeated in discussion about the implication of results.  
  3. The discussion section described the meaning of NLR briefly, that was a repeat at introduction section but did not sufficiently describe the meaning and impact of NLR in these patients population. The discussion session should be more detailed and targeted to the results of the draft. This the article about NLR, but the authors even did not describe the mean+SD, or range of NRL values in the total population, as well as high and low NLR subgroups.

Author Response

We appreciated the excellent and valuable comments.

We did our best to make a satisfactory revision to your requests.

We used "track changes" in Microsoft Word and the page in response letter is version using the track change function.

We attach the response letter and hope that our revised manuscript is satisfactory for you.

Thank you.

Reviewer: 3

This article deals with an interesting topic evaluating the prognostic value of the neutrophil-to-lymphocyte ratio(NLR) before and after radiotherapy for anaplastic thyroid cancer. Because NLR has been reported as an important systemic hematologic marker that could be used in the prediction of the outcome of various cancers, and it is not applied to very aggressive and fatal anaplastic thyroid cancer. While there have been reported that higher NLR is associated with increased tumor size and aggressiveness in differentiated thyroid cancer.

Overall, the description of results and discussion is inconsistent and very confusing. Authors have tried to demonstrate that higher NRL group has poorer outcome in general regardless of applying different therapeutic options. They showed that significantly improved overall survival in those with lower NLR (<3.47) than higher NLR group for both pre- and post-therapy data in a total group. (pre-therapy in Figure 2 and post-therapy, in Figure 4). In subgroup analysis, this was only confirmed in the group those underwent radiotherapy after surgery only. They showed that changes of NLR after RT did not have a statistically significant impact on the prediction of overall survival.

Most importantly, the authors stated that higher baseline NLR was associated with improved survival (page 5 line 181-182) and even in the discussion section (page 9. Line 243-245). What happened? Higher NLR Authors described completely oppositely from those data and repeated in discussion about the implication of results. 

Answer] The authors do appreciate the reviewer’s valuable comments. We really apologize for inconsistency between data results and description and do appreciate for pointing out precisely. The result and discussion contents was modified to match our data results.

[Result, 3.3. Overall survival according to the baseline NLR, page 6]:

Figure 2 shows the Kaplan–Meier curves for OS. The high NLR was associated with poorer survival (p = 0.002).

[Discussion, 1st paragraph, page 10]

The high baseline NLR was significantly associated with poorer OS in both univariate and multivariate analyses for the total population.

The discussion section described the meaning of NLR briefly, that was a repeat at introduction section but did not sufficiently describe the meaning and impact of NLR in these patients population. The discussion session should be more detailed and targeted to the results of the draft.

Answer] The authors do appreciate your helpful comments. We added meaning and role of inflammatory cells and NLR in thyroid cancer. We also mentioned whether NLR could discriminated ATC from differentiated thyroid cancer and why a tumor size was not associated with NLR in our study. Moreover, we discussed the need for future study whether NLR can predict the response of immunotherapy in thyroid cancer, because high NLR are known to be associated with poorer outcome in the patients with immune check point inhibitor in other solid cancer. Lastly, we added limitation of this study not performing immunohistochenical staining or Fluorescence-activated cell sorting analyses to evaluate tumor associated neutrophil or lymphocyte subset in cancer tissue.

[Discussion, 2nd, 3rd and 4th paragraph, page 10-11]

Although high NLR have been reported to be associated with poor survival in many solid cancer, the association between ATC and NLR has not been reported yet. The role of the inflammatory cells in the development and progression of thyroid cancer have been discussed in previous studies [17-19]. To date, it is known that chronic lymphocytic thyroiditis in PTC induce improved prognosis and lymphocytes are less in the patients with poorly differentiated thyroid cancer (PDTC) and ATC than in PTC patients suggesting lymphocyte have protective role in thyroid cancer [18]. The monocytes accelerate tumor progression and suppress normal immune system and low lymphocyte-to-monocyte ratio are associated with poor survival in ATC [15,17]. The neutrophil secrets highly reactive metabolites and eliminate pathogen. However, these metabolites can induce mutation or proliferation of the cancer cell [18]. Our study showed high NLR was associated with poor survival before and after RT in ATC, which is consistent results with the previous studies considering the role of each inflammatory cells. As we mentioned in introduction, it was reported that high NLR was associated with increased tumor size, metastasis and disease free survival in patients with differentiated thyroid cancer [7,8].

As the tumor size and staging increases, higher amounts of tumor derived or tumor induced factors are likely to be released into the periphery of cancer thus increasing the circulating neutrophil amount [20]. In our study, tumor size was not associated with NLR. PTC is indolent form of cancer, grows slowly and NLR value is also lower compared to ATC [10]. This means inflammation may be less important in the initiation of carcinogenesis in differentiated cancer, less affecting systemic inflammation [8]. The NLR may increase as the size increases provoking systemic inflammation gradually in PTC. On the other hand, ATC is aggressive and grows fast inducing inflammatory process in early phase of carcinogenesis. Therefore, NLR was thought not to be associated with tumor size. Furthermore, one study reported that NLR can discriminate aggressive forms of thyroid cancer from well differentiated cancer [10]. In this previous study, NLR of PTC, 1.57 (range, 0.28-16.29) could discriminate PDTC and ATC from PTC and the cut-off value of LNR for discriminating PDTC from ATC was 3.8. Considering the NLR of our study population was 2.87 (range, 0.94-22.51), it is higher value compared to PTC patients in previous study. However, only 14 out of 40 our study population had NLR value same or over than 3.8. When the cut off value was set to 3.8, the sensitivity was 45.16% and specificity was 89% for death in our study population. This means even in ATC patients, NLR over 3.8 represents high mortality. Considering PDTC is as aggressive as ATC, LNR over 3.8 can discriminate aggressive type of thyroid cancer and poor survival. Studies in larger patient cohorts are warranted to define uniform NLR cut-off value for thyroid cancer.

High NLR is also associated with poorer outcome in the patient with immune checkpoint inhibitor in other solid cancer [21]. Circulating neutrophil counts are associated with tumor infiltrating neutrophil [22]. Therefore, decreased circulating lymphocyte are associated with reduced lymphocyte in tumor tissue resulted in less chance to interact with immune check point inhibitor. Further intervention studies are needed to confirm that NLR can predict the response of immunotherapy in thyroid cancer.

[Discussion, 7th paragraph, page 11]

Third, we could not perform additional immunohistochemical staining to find tumor associated neutrophil or flow cytometry to evaluate lymphocyte subtype due to lack of available tissue specimen. Therefore, the difference of neutrophil and lymphocyte infiltration in ATC between low NLR and high NLR groups were not evaluated. However, we focused the association between blood NLR and OS of the patients with ATC predicting blood NLR could reflect tumor tissue NLR. Further studies are needed to confirm the consistency of NLR in blood and tissue.

This the article about NLR, but the authors even did not describe the mean+SD, or range of NRL values in the total population, as well as high and low NLR subgroups.

Answer] Thank you very much for helpful comments. We actively reflect your comments and added the results of baseline NLR in total population to result “3.2 Optimal cutoff value for the NLR and clinicopathological characteristics according to NLR” section and subgroup NLR data to Table 1 and Table 2. Furthermore, the value of post-RT-NLR was added to result “2.5. Association between NLR after RT and OS” section.

[Result, 3.2. Optimal cutoff value for the NLR and clinicopathological characteristics according to NLR, page 5]

The median and IQR of NLR in total population was 2.87 (1.50-5.28).

[Result, 3.2. Optimal cutoff value for the NLR and clinicopathological characteristics according to NLR, 2nd paragraph, page 5]

Clinicopathological characteristics of patients according to NLR are shown in Table 2. The NLR was 1.99 (1.28-2.67) in low NLR group and 6.33 (4.59-11.11) in high LNR group (p<0.001).

[Result, 2.5. Association between NLR after RT and OS, 1st paragraph, page 8]

The median and IQR of post-RT-NLR was 6.60 (2.75-9.81).

[Result, Table 1 and Table 2, page 4-5]

Table 1. Baseline characteristics of patients who received RT with or without surgery for       anaplastic thyroid carcinoma.

OP + RT ± Systemic Tx

(n = 19)

RT ± Systemic Tx

(n = 21)

p-value

Age (years)

65.1 (57.8–72.2)

70.8 (58.1–78.6)

0.383

Sex

0.220

Male

9 (47.4)

6 (28.6)

Female

10 (52.6)

15 (71.4)

Tumor size (cm)

4.8 (3.1–5.4)

5.3 (4.2–6.5)

0.245

Stage

0.012

IVA & IVB

13 (68.4)

6 (28.6)

IVC

6 (31.6)

15 (71.4)

Total dose (EQD210, Gy)

<0.001

<60.0

7 (36.9)

19 (90.5)

≥60.0

12 (63.1)

2 (9.5)

Systemic treatment

Cytotoxic chemotherapy

5 (26.32)

9 (42.86)

0.273

TKI

3 (15.8)

7 (33.3)

0.201

Multimodal therapy

19 (100)

13 (61.9)

0.003

Baseline ALC (µL)

2120 (1740–2363)

1810 (1310–2380)

0.173

Baseline ANC (µL)

4220 (2970–5803)

6598 (5814–11729)

0.017

Baseline NLR

2.24 (1.4-3.1)

4.95 (2.54-6.60)

0.013

Baseline NLR (≥3.47)

3 (15.8)

12 (57.1)

0.007

Table 2. Clinicopathological characteristics according to the baseline NLR in patients with anaplastic thyroid carcinoma treated by radiotherapy.

Low NLR

(n = 25)

High NLR

(n = 15)

p-value

Age (years)

67.4 (63.1–72.2)

67.3 (57.9–73.0)

0.882

Sex

0.354

Male

8 (32.0)

7 (53.3)

Female

17 (68.0)

8 (46.7)

Tumor size (cm)

4.5 (3.3–5.5)

5.4 (4.4–6.5)

0.145

Stage

0.165

IVA & IVB

14 (56.0)

5 (33.3)

IVC

11 (44.0)

10 (66.7)

Total dose (EQD210, Gy)

0.123

<60.0

14 (56.0)

12 (80.0)

≥60.0

11 (44.0)

3 (20.0)

Surgery

16 (64.0)

3 (20.0)

0.007

Systemic treatment

Cytotoxic chemotherapy

7 (28.0)

7 (46.7)

0.231

TKI

6 (24.0)

4 (26.7)

0.850

Multimodal therapy

20 (80.0)

12 (80.0)

1.000

Baseline NLR

1.99 (1.28-2.67)

6.33 (4.95-11.11)

<0.001

Baseline ALC (µL)

2218 (1890–2460)

1480 (1222–1853)

<0.001

Baseline ANC (µL)

4188 (2680–5802)

10494 (7677–14060)

<0.001

Thank you very much for the helpful comments and suggestions. We did our best to address the reviewer’s requests and hope that it would be the satisfied version for the publication.

Round 2

Reviewer 1 Report

I have appreciated the satisfactory responce to my requests accepting the manuscript in this form 

Thank you. 

Reviewer 3 Report

The authors corrected false statements and revised the manuscript, per the reviewer's comments. No more opinion.